# Comprehensive Assessment of Left Intraventricular Hemodynamics Using a Finite Element Method: An Application to Dilated Cardiomyopathy Patients

Pamela Franco [1,2,3], Julio Sotelo [1,3,4], Cristian Montalba [1,3,5], Bram Ruijsink [6,7], Eric Kerfoot [6], David Nordsletten [8,9], Joaquín Mura [3,10], Daniel Hurtado [3,11,12] and Sergio Uribe [1,3,5,*]

1 Biomedical Imaging Center, School of Engineering, Pontificia Universidad Católica de Chile, Santiago 7820436, Chile; pafranco@uc.cl (P.F.); julio.sotelo@uv.cl (J.S.); cristian.montalba@uc.cl (C.M.)
2 Electrical Engineering Department, School of Engineering, Pontificia Universidad Católica de Chile, Santiago 7820436, Chile
3 Millennium Nucleus for Cardiovascular Magnetic Resonance, Santiago 7820436, Chile; joaquin.mura@usm.cl (J.M.); dhurtado@ing.puc.cl (D.H.)
4 School of Biomedical Engineering, Universidad de Valparaíso, Valparaíso 2362905, Chile
5 Radiology Department, School of Medicine, Pontificia Universidad Católica de Chile, Santiago 833023, Chile
6 Guy's and St Thomas' NHS Foundation Trust, London SE1 7EH, UK; jacobus.ruijsink@kcl.ac.uk (B.R.); eric.kerfoot@kcl.ac.uk (E.K.)
7 School of Imaging Sciences and Biomedical Engineering, Faculty of Science of Life and Medicine, King's College London, London WC2R 2LS, UK
8 Division of Biomedical Engineering and Imaging Sciences, Department of Biomedical Engineering, King's College London, London WC2R 2LS, UK; nordslett@umich.edu
9 Department of Biomedical Engineering and Cardiac Surgery, University of Michigan, North Campus Research Center, Ann Arbor, MI 48109, USA
10 Department of Mechanical Engineering, Universidad Técnica Federico Santa María, Valparaíso 2390123, Chile
11 Structural and Geotechnical Engineering Department, School of Engineering, Pontificia Universidad Católica de Chile, Santiago 7820436, Chile
12 Institute for Biological and Medical Engineering, School of Engineering, Medicine and Biological Sciences, Pontificia Universidad Católica de Chile, Santiago 7820436, Chile
* Correspondence: suribe@uc.cl

**Abstract:** In this paper, we applied a method for quantifying several left intraventricular hemodynamic parameters from 4D Flow data and its application in a proof-of-concept study in dilated cardiomyopathy (DCM) patients. In total, 12 healthy volunteers and 13 DCM patients under treatment underwent short-axis cine b-SSFP and 4D Flow MRI. Following 3D segmentation of the left ventricular (LV) cavity and registration of both sequences, several hemodynamic parameters were calculated at peak systole, e-wave, and end-diastole using a finite element approach. Sensitivity, inter- and intra-observer reproducibility of hemodynamic parameters were evaluated by analyzing LV segmentation. A local analysis was performed by dividing the LV cavity into 16 regions. We found significant differences between volunteers and patients in velocity, vorticity, viscous dissipation, energy loss, and kinetic energy at peak systole and e-wave. Furthermore, although five patients showed a recovered ejection fraction after treatment, their hemodynamic parameters remained low. We obtained several hemodynamic parameters with high inter- and intra-observer reproducibility. The sensitivity study revealed that hemodynamic parameters showed a higher accuracy when the segmentation underestimates the LV volumes. Our approach was able to identify abnormal flow patterns in DCM patients compared to volunteers and can be applied to any other cardiovascular diseases.

**Keywords:** 4D flow MRI; flow quantification; finite elements; left ventricle; dilated cardiomyopathy

## 1. Introduction

Dilated cardiomyopathy (DCM) is more common than non-ischemic cardiomyopathy and leads to left ventricular dilation and systolic and diastolic dysfunction [1,2]. The

process that alters the heart's size, geometry, and function is associated with increased hemodynamic demands, which cause abnormal mechanical stress in the muscle [3,4]. The progression is associated with an incremented risk of heart failure and sudden cardiac death [5]. However, poor survival and high mortality rate reveal that effective treatment of DCM-related heart failure remains challenging. Pharmacological and resynchronization therapies have improved DCM treatment by halting disease progression and leading to reverse remodeling [1].

The preferred imaging technique for assessing the heat in DCM patients is cardiovascular magnetic resonance (CMR). CMR allows the acquisition of anatomical, cine, and velocity images, including 4D Flow MR [5–8].

4D Flow allows a qualitative and quantitative analysis of several hemodynamic parameters. It has been applied extensively in the great vessels, particularly in the aorta [9–11] and in the left ventricle (LV) for assessing intraventricular flow in some cardiovascular diseases [12–19]. Previous studies have demonstrated that lower kinetic energy values in diastole are associated with the deterioration of ventricular filling, induced by morphological alteration commonly found in Fontan patients, mitral regurgitation, and LV dysfunction or remodeling [13–15]. Additionally, turbulent kinetic energy has shown a stronger association with the ventricle's remodeling in patients with Tetralogy of Fallot and higher values in DCM patients compared with normal subjects [16]. Vortex formation has been studied qualitatively (vortex size and location) and quantitatively (Lagrangian Coherent Structures and the curl of velocity) [17–20]. These studies suggest that parameters associated with the 3D intraventricular flow may be critical for LV filling and ejection and could be relevant to the development of dilation, dysfunction, and prognosis in patients with heart diseases. While these measures have a potential role in describing intraventricular flow, the difficulties of implementing them have led to the analysis of only a few combinations of these parameters in a single cohort of patients [13–15,17,19].

Due to the multidirectional velocity data, although impressively comprehensive, it may need to be supplemented by more selective flow imaging at high temporal and spatial resolutions or computational fluid dynamics simulation. Reaching conclusions regarding small-scale methodology, which comprehensively describes the characteristics of intraventricular flow, could improve the use of intraventricular 4D Flow for clinical research and potential translation to clinical settings.

In this work, we adapted a method for quantifying 4D Flow in the aorta [9–11,20]. We modified the methodology applied in the left ventricle to obtain several hemodynamic parameters from a single segmentation from a 4D Flow dataset and cine MRI. To show the applicability of this approach, we performed a proof-of-concept study in which we applied the method in a small cohort of DCM patients to find which parameters were different from volunteers. We obtained three-dimensional hemodynamic parameters, including kinetic energy, vorticity, helicity density, viscous dissipation, and energy loss [9,13,21–25].

## 2. Materials and Methods

### 2.1. Population

A total of 12 healthy volunteers (HV), mean age 40.8 years (range 27–55 years), and 13 DCM patients, mean age 48.7 years (range 29–62 years), matched according to age and gender, were included in this research. Demographical and clinical data are described in Table 1. At the time of diagnosis, DCM was defined as the presence of symptoms and signs of heart failure with echocardiographic signs of ventricular enlargement and systolic myocardial dysfunction in the absence of hypertension, valve diseases, or significant coronary artery diseases sufficient to cause global systolic impairment, by the definition of the European Society of Cardiology [26]. Our DCM cohort all received treatment with an improved LV ejection fraction (range 51–66%) and LV volume indices at CMR imaging. All patients received standard guideline-directed treatment for DCM following the 2008 heart failure guidelines from the European society of cardiology. The details of treatments were not available, as our center is the referral center for several clinics for cardiac CMR. The

HV had normal electrocardiograms and echocardiographic examinations without valvular or ventricular dysfunction. All subjects participated under informed consent, with data collection approved by the Regional Ethics Committee, South East London, UK (REC, 12/LO/1456).

**Table 1.** Demographical and clinical data for healthy and DCM patients. All quantitative data are expressed as the median (range). HR: Heart Rate, EF: Ejection Fraction, LVSV: Left Ventricle Stroke Volume, CO: Cardiac Output, LVEDV: Left Ventricle End-Diastolic Volume, and Left Ventricle End-Systolic Volume. * indicates statistically significant differences ($p < 0.05$).

| | | | | DCM Group | | |
|---|---|---|---|---|---|---|
| | **HV** | **DCM** | ***p*-Value** | **LVEF $\geq$ 50 (Complete-Responders)** | **LVEF < 50 (Non-Responders)** | ***p*-Value** |
| **N** | 12 | 13 | | 5 | 8 | |
| **Age (years)** | 39 (27,55) | 51 (29,62) | 0.060 | 40 (29,62) | 53 (44,58) | 0.502 |
| **Gender (female:male)** | 5:7 | 6:7 | 0.821 | 3:2 | 3:5 | 0.429 |
| **Weight (kg)** | 68 (50,111) | 83 (43,116) | 0.213 | 90 (72,116) | 72.5 (43,95) | 0.071 |
| **Height (cm)** | 173 (163,188) | 168 (155,178) | 0.203 | 168 (163,178) | 166.5 (155,175) | 0.454 |
| **HR (bpm)** | 64 (58,78) | 65 (56,101) | 0.743 | 65 (56,101) | 67.5 (57,89) | 0.698 |
| **EF (%)** | 62.7 (54,69) | 46 (29,66) | <0.001 * | 55 (51,66) | 44 (29,48) | 0.002 * |
| **LVSV (mL)** | 95.5 (66.3,122.9) | 62 (53,132.1) | 0.039 * | 61 (53,89) | 79 (55,132,1) | 0.183 |
| **CO (L/min)** | 6.4 (4.8,7.9) | 6.1 (4.4,7.9) | 0.327 | 6.3 (5.2,7.9) | 5.9 (4.4,7.7) | 0.524 |
| **LVEDV (mL)** | 153 (105.6,197.1) | 199 (125,364.2) | 0.015 * | 187 (151,201) | 219.5 (125,364.2) | 0.050 * |
| **LVESV (mL)** | 51 (39,88) | 92 (37,232.1) | 0.004 * | 75 (37,92) | 125 (68,232.1) | 0.045 * |

### 2.2. Data Acquisition

Multi-slice 2D cine balanced steady-state free precession (b-SSFP) and 4D Flow MRI data were acquired in all subjects using a clinical 1.5 T MT Scanner (Philips Achieva, Philips Medical Systems, Best, The Netherlands). During the MRI examination, multi-slice b-SSFP was used to acquire short-axis morphological images in 40 frames with 8 mm slice thickness, using retrospective cardiac gating. Acquisition parameters were echo time (TE) 1.4 ms, repetition time (TR) of 2.8 ms, flip angle (FA) of 60° and acquired and reconstructed pixel sizes were $2.47 \times 2.53$ mm$^3$ and 1.45 mm$^2$, respectively. 4D Flow MRI data were acquired during free-breathing with MR parameters, as follows: TW of 2.3 ms, TR of 4.7 ms, FA of 6°, velocity encoding of 130 cm/s, and spatial resolution (acquired and reconstructed) $2.5 \times 2.5 \times 2.5$ mm$^3$. These settings gave a temporal resolution of 58 ms. After the acquisition, the 4D Flow MRI data were reconstructed into 24 cardiac phases on the MRI system.

### 2.3. Data Analysis

The 4D Flow MRI datasets were processed using an-house MATLAB library (The MathWorks Inc., Natick, MA, USA), which included the registration of the b-SSFP cine and 4D Flow MR images, interpolation of the b-SSFP images, segmentation of the LV, and generation of the finite element mesh (Figure 1).

The Eidolon software was used to perform the registration between the multi-slice b-SSFP and the 4D Flow MRI (King's College London, London, UK) [27]. To obtain a smooth tetrahedral mesh, we doubled the number of slices in the b-SSFP images by using a cubic interpolation of values at neighboring grid points in each respective dimension, obtaining a final voxel size of $1.43 \times 1.43 \times 4.04$ mm$^3$. LV endocardium was automatically segmented throughout all cardiac phases in the short-axis cine b-SSFP images, using the image analysis software Segment v2.2R6410 (Medviso AB, Lund, Sweden) [28–30]. The segmentation was visually inspected and manually corrected if needed. Segmentations c were then used to generate a binary mask. Afterward, we created a tetrahedral mesh using the iso2mesh MATLAB Toolbox [31]. Once the mesh was constructed, we computed the velocity vector at each mesh node from the 4D Flow datasets using a cubic interpolation. 3D maps of vorticity, helicity density, viscous dissipation, energy loss, and kinetic energy fields were then calculated using a previously published finite element approach [9–11,25]. The description of the equations used to calculate each hemodynamic parameter is presented

in Table S1. The parameters were averaged at peak systole, e-wave, and end-diastole using one timeframe before and after to reduce noise in the data.

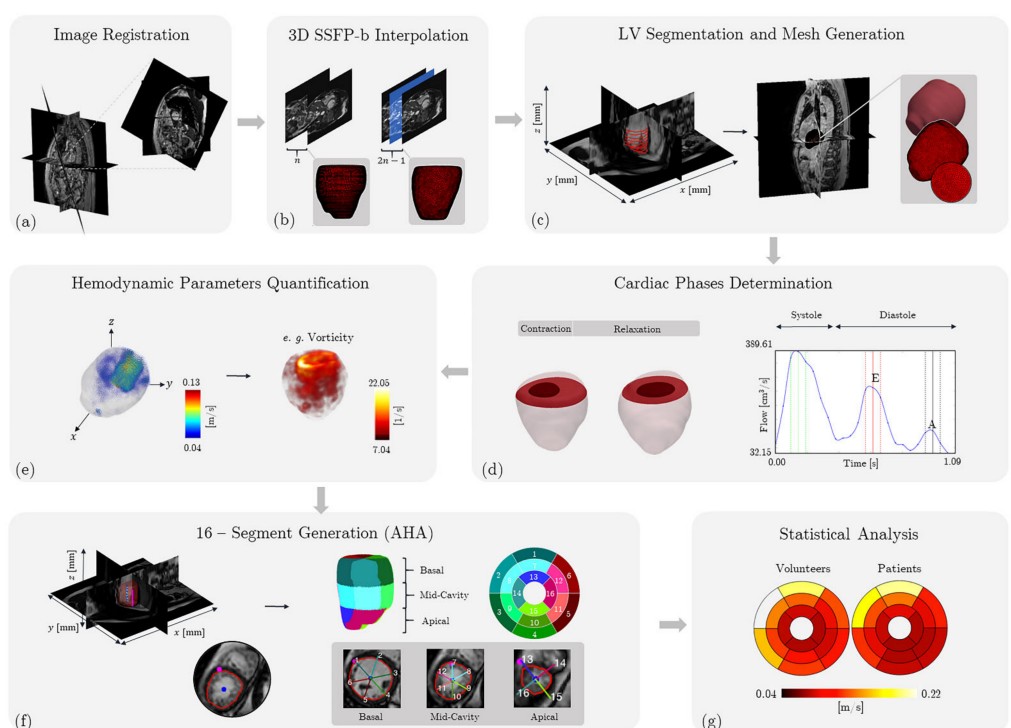

**Figure 1.** Schematic description of the quantification process. (**a**) First, we performed registration of the 4D Flow with the b-SSFP images. (**b**) Second, we doubled the number of slices in the b-SSFP images, (**c**) then the LV segmentation and tetrahedral mesh were generated. (**d**) We estimated the cardiac phases under study (**e**) and we transferred the velocity information at each node of the mesh from the 4D Flow MRI datasets using cubic interpolation. (**f**) Then, we calculated hemodynamic parameters under study. (**g**) Finally, the mean values of hemodynamic parameters were included in a bullseye plot to compare volunteers and DCM patients.

### 2.4. Local Hemodynamics

A 16-segment model was used to divide the LV. In contrast to a standardized nomenclature, a minor adjustment was made [26]. Due to the generally intricate shape of the apical region of the LV, region 17 was excluded from our analysis. Accordingly, LV mesh was divided into 16 segmentations. The centerline of the LV was calculated automatically by detecting the centroid of the LV contour in each slice and connected to create a line. To determine the three sections of the LV, we divided the centerline into three equal parts perpendicular to the long axis of the heart. An additional point was then manually placed at the junction between the right ventricular free wall and the interventricular septum on the LV. Based on these positions, landmarks were uniformly distributed along the boundaries. Each section was then partitioned into six segments of 60° each on basal and mid-cavity sections and four segments of 90° each on apical section (Figure 1f). Finally, for visualization purposes, we used the scientific software ParaView version 5.3.0 (Kitware, Clifton Park, NY, USA).

### 2.5. Statistical Analyses

Normal distribution in population demographics was evaluated using the Shapiro-Wilk test. Differences between groups for continuous parameters were assessed by a Student *t*-test if they presented a normal distribution, and the Mann-Whitney *U* test otherwise. The $\chi^2$ test was applied for categorical variables, which were reported as percentages. A *p*-value < 0.05 was considered statistically significant. The statistical

analyses were performed using GraphPad Prism version 6.0.1 (GraphPad Software Inc., San Diego, CA, USA).

These data were displayed in box-whisker and bullseye plots for global and local analyses, respectively. Additionality, a correlation matrix-based hierarchical clustering method was introduced to extract multiple correlation patterns from hemodynamic parameters. This method can effectively identify highly correlated data. The results are described with a tree structure plot called a dendrogram. The present study used Pearson's correlation method to measure the similarity between hemodynamic parameters [32].

Furthermore, a sensitivity study was performed by looking at changes in the hemodynamic parameters subjected to the LV segmentation changes. We increased and decreased the size of the LV cavity from the first segmentations by moving the segmentation contour in 0.5 to 2 pixels of the b-SSFP image, equivalent to 0.72 to 2.89 mm. We compared the results with the original LV segmentation's respective mean value at each cardiac phase studied. We used the Kruskal-Wallis test to compare the variables across the different LV segmentation, with a *p*-value < 0.05 indicating statistical significance. The significance level was adjusted by using Dunn's test correction.

To assess the inter-observer agreement, data were analyzed by two independent observers, one with three years of experience in MR LV quantification and the other a medical technologist with no previous experience in this field. In addition, re-analyzed images with a 1-month interval to evaluate the intra-observer reproducibility. Inter- and intra-observer reproducibility were analyzed using Bland-Altman plots, and the results are shown in the Supplementary Materials.

## 3. Results

### 3.1. Study Population

There were no significant differences in age and heart rate (Table 1). However, ejection fraction and stroke volume were lower in DCM patients than volunteers, while end-diastolic and end-systolic volumes were larger. These changes indicated that the LV in DCM patients was enlarged and its cardiac function was reduced, which is consistent with the pathological characteristics of DCM [1,2]. Additionally, eight patients still showed significantly impaired systolic function at CMR's time (non-responders), and five patients showed a complete response to treatment (complete responders). Between DCM groups, complete- vs. non-responders, there were differences in ejection fraction, end-diastolic volume, and end-systolic volume.

### 3.2. Global Hemodynamics

Assessment of global hemodynamic parameters is shown in Figure 2 and Table 2. Volunteers showed higher hemodynamics values than patients at peak systole and e-wave, except for helicity density. Remarkably, hemodynamic parameters in complete responder DCM patients remained low compared to volunteers. We found statistical differences between HV and DCM patients: non- and complete responders at peak systole and e-wave in velocity, vorticity, viscous dissipation, energy loss, and kinetic energy. In all cases, *p*-values were lower or equal to 0.005. There were no statistical differences in the parameters at end-diastole. Furthermore, we did not find statistical differences between DCM groups. In addition, ROC curves showed that previous parameters discriminated between HV and DCM patients (Figure S1).

The total computational time used to process the data, once the multi-slice b-SSFP was segmented and registered, varied between 30–40 s for one cardiac phase, using a standard computer (3.4 GHz Intel $^{\circledR}$ Core i7$^{TM}$, 16 GB RAM).

Hierarchical cluster analysis (Figure 3) provides an alternative method for reliable identification of correlation between ejection fraction and hemodynamics parameters from 4D-flow MRI. According to their similarities, they are classified into two clusters identified at peak-systole, e-wave, and end-diastole. At peak-systole and e-wave, cluster 1 (black): helicity density; and cluster 2 (red): ejection fraction, energy loss, vorticity, viscous

dissipation, velocity, and kinetic energy. Finally, at end-diastole, cluster 1 (red): helicity density and ejection fraction, and cluster 2 (red): energy loss, viscous dissipation, vorticity, velocity, and kinetic energy. This means that ejection fraction correlates with all parameters except helicity density at peak systole and e-wave.

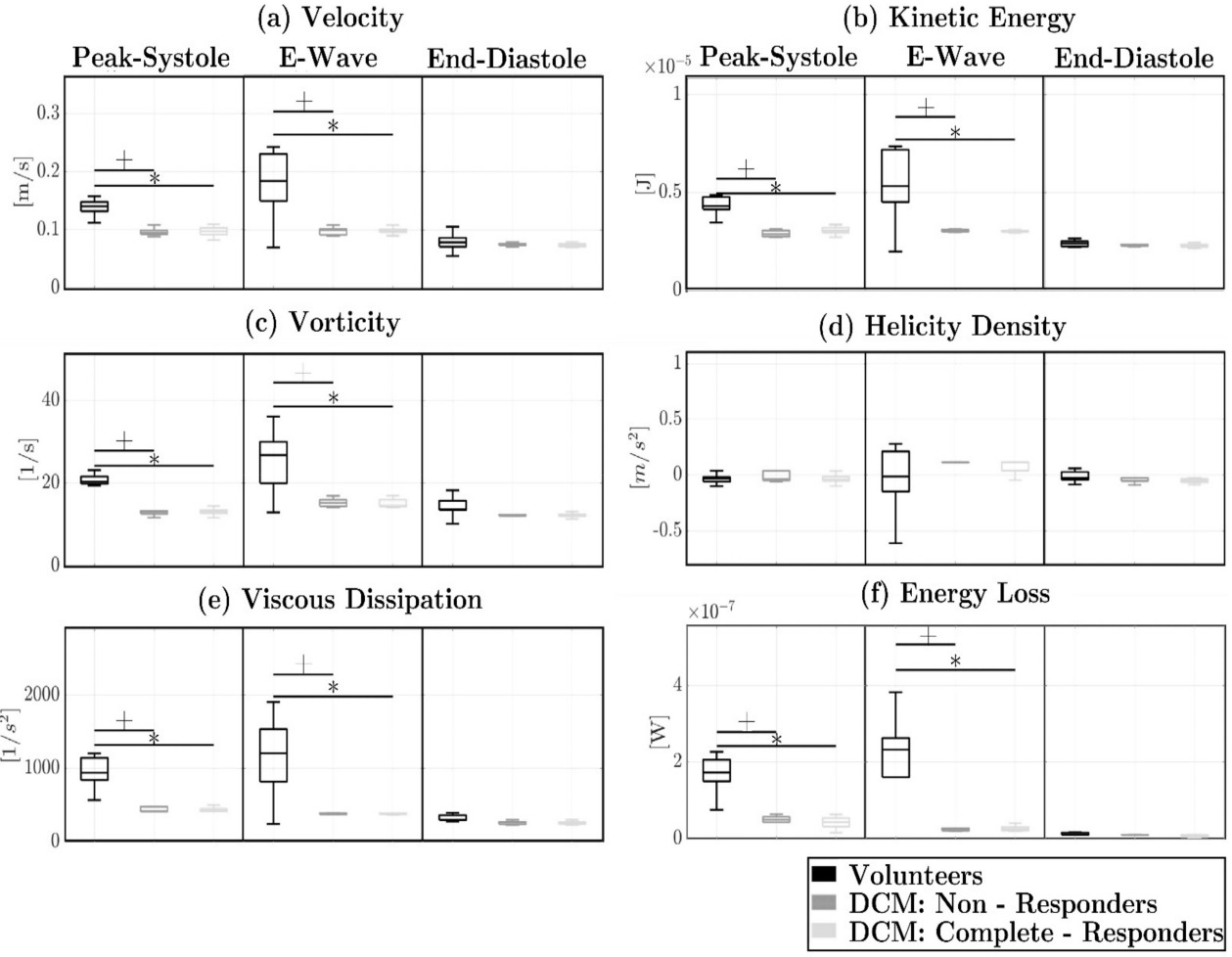

**Figure 2.** Box whisker plots for hemodynamic parameters (**a–f**) in the entire LV cavity of HV and DCM patients groups at peak systole, e-wave, and end-diastole. On each box, the central mark is the median, the bottom and top edges of the box are the 25th and 75th percentiles, respectively, and the whiskers extend to the most extreme data points not considered outliers. *,+ Indicates statistically significant differences ($p < 0.05$).

### 3.3. Sensitivity Study, Intra-, and Inter-Observer Reproducibility

Figure 4 shows a sensitivity analysis at peak-systole. The relative error in LV cardiac volumes and helicity density did not show significant differences between groups across LV segmentation. There were significant differences for some segmentation for the other parameters, particularly for the velocity magnitude, energy loss, and kinetic energy. The hemodynamic parameters showed a relative error proportional to the dilatation or erosion of the contour in the segmentation. When the segmentation was dilated or eroded 1 pixel or less, the relative error differences, with respect to the original segmentation for volunteers and DCM patients, were: velocity magnitude (9.03%, 6.78%), vorticity magnitude (5.49%, 2.74%), helicity density (12.11%, 13.98%), viscous dissipation (6.31%, 3.34%), energy loss (3.59%, 5.89%), and kinetic energy (7.66%, 6.09%). Similar results were obtained at e-wave and end-diastole, as shown in Figures S2 and S3. Those errors were more significant, particularly when the segmentation was dilated or eroded by more than 1 pixel. Helicity density and energy loss showed greater dependency on the segmentation error.

Regarding reproducibility, there was an excellent agreement of inter- and intra-observer analysis of global hemodynamic parameters. Details are given in the Appendix A.

**Table 2.** Global hemodynamics data for HV, complete, and non-responders DCM patients. All quantitative data are expressed as the mean ± standard deviation. *,+ Indicates statistically significant differences ($p < 0.05$).

| | | DCM Group | | *p*-Value | |
|---|---|---|---|---|---|
| | HV | Complete-Responders | Non-Responders | HV vs. Complete-Responders | HV vs. Non-Responders |
| **Peak-systole** | | | | | |
| **Velocity (m/s)** | 0.140 ± 0.014 | 0.099 ± 0.007 | 0.096 ± 0.008 | <0.001 * | <0.001 + |
| **Kinetic Energy (µJ)** | 43.722 ± 4.592 | 29.335 ± 1.917 | 31.288 ± 2.044 | <0.001 * | <0.001 + |
| **Vorticity (1/s)** | 20.306 ± 2.075 | 12.934 ± 0.814 | 13.331 ± 1.251 | <0.001 * | <0.001 + |
| **Helicity Density (m/s²)** | −0.042 ± 0.004 | 0.036 ± 0.161 | −0.077 ± 0.139 | 0.125 | 0.417 |
| **Viscous Dissipation (1/s²)** | 970.840 ± 412.093 | 412.093 ± 61.107 | 421.080 ± 54.870 | <0.001 * | <0.001 + |
| **Energy Loss (ηW)** | 173.080 ± 39.387 | 35.284 ± 14.144 | 47.734 ± 12.935 | <0.001 * | <0.001 + |
| **E-wave** | | | | | |
| **Velocity (m/s)** | 0.187 ± 0.059 | 0.097 ± 0.004 | 0.099 ± 0.014 | 0.007 * | <0.001 + |
| **Kinetic Energy (µJ)** | 5.567 ± 1.810 | 3.008 ± 0.074 | 3.005 ± 0.423 | 0.007 * | <0.001 + |
| **Vorticity (1/s)** | 26.309 ± 7.895 | 14.633 ± 0.755 | 14.931 ± 1.963 | 0.005 * | <0.001 + |
| **Helicity Density (m/s²)** | 0.106 ± 0.441 | 0.077 ± 0.065 | 0.056 ± 0.119 | 0.907 | 0.785 |
| **Viscous Dissipation (1/s²)** | 1208.091 ± 574.696 | 393.994 ± 26.632 | 370.314 ± 80.785 | 0.007 * | <0.001 + |
| **Energy Loss (ηW)** | 217.440 ± 126.751 | 28.408 ± 6.340 | 24.329 ± 11.540 | 0.005 * | <0.001 + |
| **End-Diastole** | | | | | |
| **Velocity (m/s)** | 0.075 ± 0.015 | 0.077 ± 0.005 | 0.073 ± 0.002 | 0.796 | 0.673 |
| **Kinetic Energy (µJ)** | 2.291 ± 0.423 | 2.359 ± 0.165 | 2.266 ± 0.063 | 0.739 | 0.871 |
| **Vorticity (1/s)** | 13.629 ± 2.255 | 12.444 ± 0.633 | 12.203 ± 0.552 | 0.273 | 0.099 |
| **Helicity Density (m/s²)** | −0.025 ± 0.084 | −0.082 ± 0.064 | −0.027 ± 0.096 | 0.201 | 0.979 |
| **Viscous Dissipation (1/s²)** | 300.577 ± 89.250 | 289.131 ± 54.300 | 256.496 ± 16.779 | 0.795 | 0.188 |
| **Energy Loss (ηW)** | 11.162 ± 7.191 | 10.459 ± 7.757 | 8.875 ± 1.896 | 0.859 | 0.395 |

### 3.4. Local Hemodynamics

Figure 5 shows the bullseye plots of the hemodynamic parameters for volunteers and DCM patients at peak systole. More areas with statistical differences were observed mainly in velocity magnitude and kinetic energy, particularly in anteroseptal, inferior, inferolateral basal, anterior, inferoseptal, inferior, inferolateral mid-cavity, anterior, septal, and lateral apical segments (all *p*-values < 0.033). Additionally, vorticity magnitude showed statistical differences in anteroseptal basal ($p = 0.045$) and inferoseptal mid-cavity ($p = 0.046$) segments. Energy loss showed statistical differences in inferoseptal ($p = 0.029$) and inferolateral ($p = 0.046$) mid-cavity segments and in anterior ($p = 0.023$), septal ($p = 0.070$), and inferior ($p = 0.077$) apical segments. Helicity density and viscous dissipation did not show statistical differences in any parcellation.

Figure S4 shows the comparison at the e-wave. Similar to peak systole, statistical differences were in velocity magnitude and kinetic energy. Statistical differences in viscous dissipation and energy loss were found in inferolateral basal and septal, anterior, and lateral apical segments (all *p*-values < 0.049). Vorticity magnitude showed statistical differences in anteroseptal basal ($p = 0.049$) and septal ($p = 0.039$) and lateral ($p = 0.039$) apical segments. Helicity density did not show statistical differences in any parcellation of the LV.

We did not find statistical differences in any segment at end-diastole.

The mean values of local hemodynamic parameters for both groups under study at peak systole, e-wave, and end-diastole are available in Tables S2–S4, respectively.

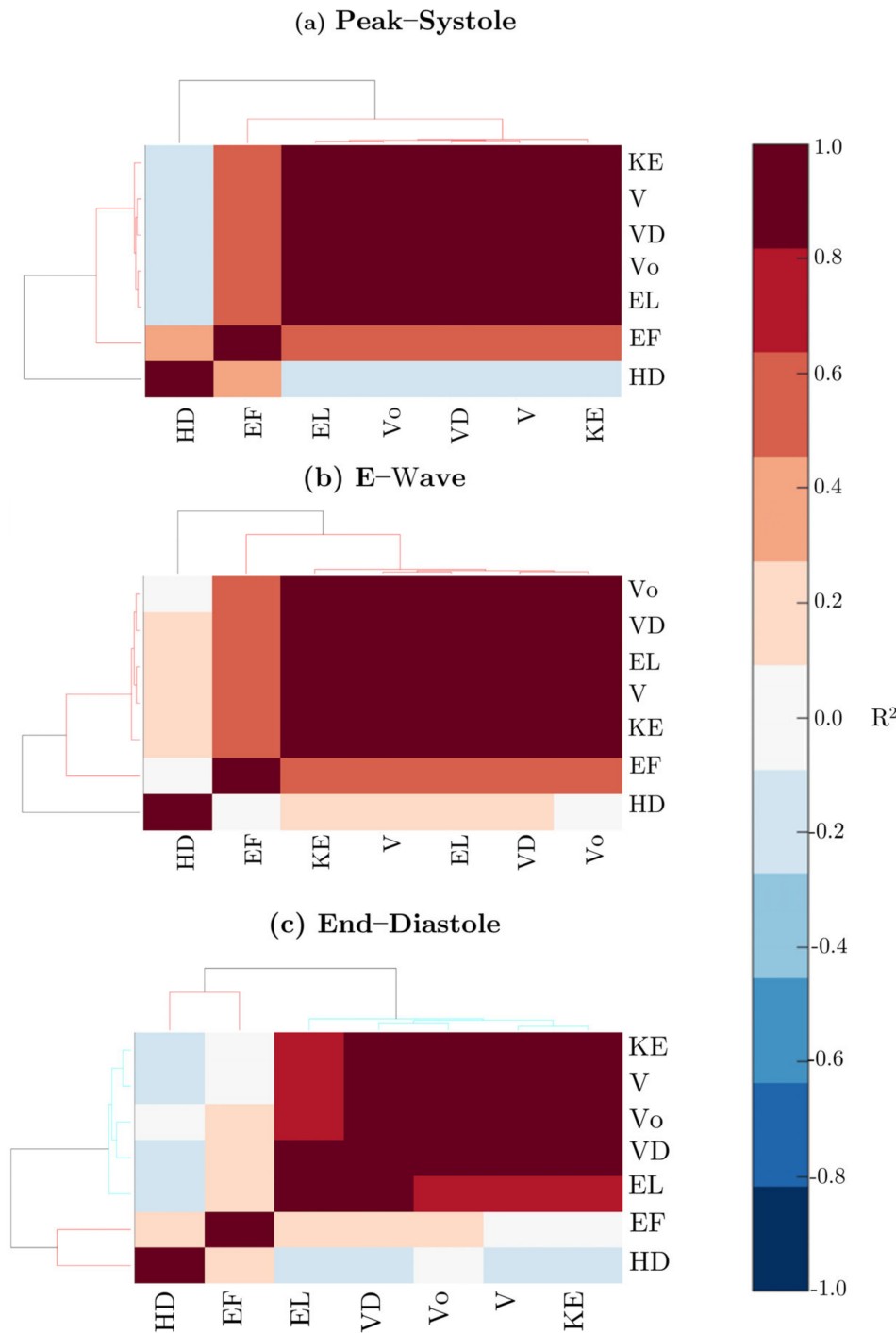

**Figure 3.** Dendrogram and hierarchical clustering results based on average linkage method for ejection fraction and hemodynamic parameters. EF: ejection fraction, V: velocity, KE: kinetic energy, Vo: vorticity, HD: helicity density, VD: viscous dissipation, and EL: energy loss.

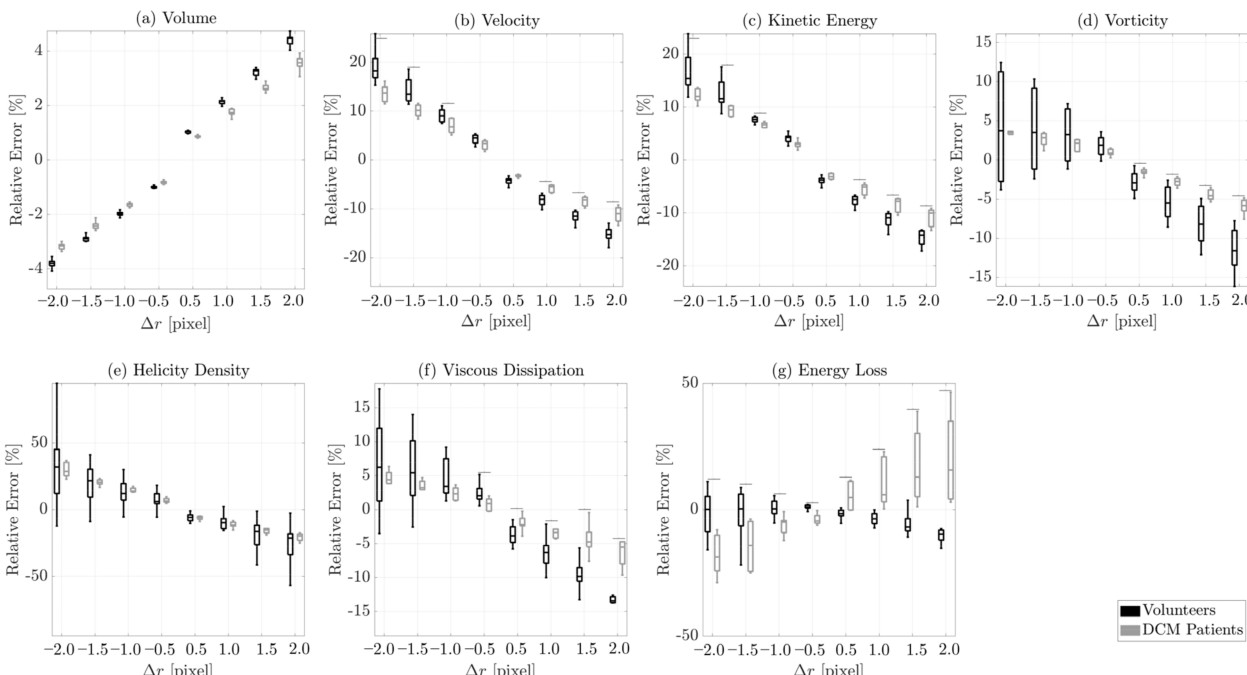

**Figure 4.** Relative error values of the volume (**a**) and each hemodynamic parameter (**b**–**g**) obtained comparing the reference segmentation with segmentations given by erosion or dilatation for each group of volunteers and patients at peak systole. * indicates statistically significant differences ($p < 0.05$).

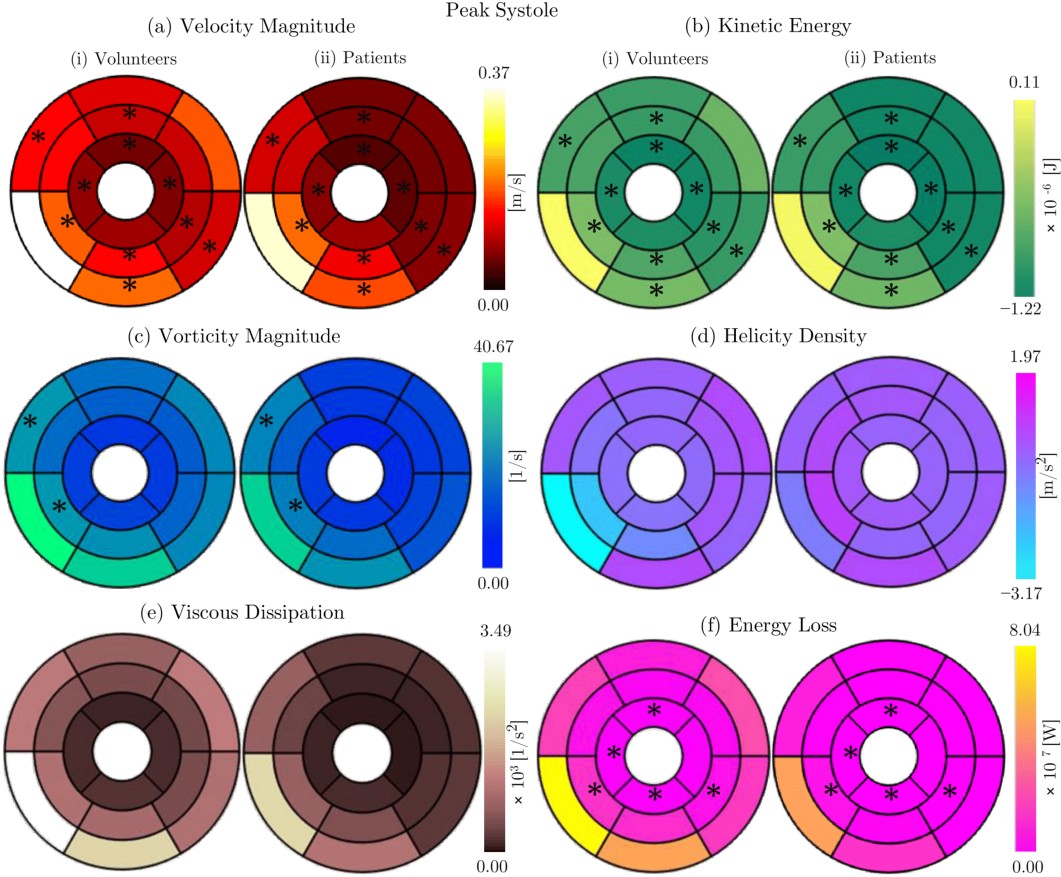

**Figure 5.** Bullseye plots of mean hemodynamic parameters (**a**–**f**) across 16 segments for volunteers (i) and patients (ii) at the peak systole. * indicates statistically significant differences ($p < 0.05$).

## 4. Discussion

We developed a method to characterize the left intraventricular hemodynamics in the LV from 4D Flow MRI using a finite element method, applied in a cohort of DCM patients. This approach estimates vorticity, helicity density, viscous dissipation, energy loss, and kinetic energy fields from a single segmentation. The hemodynamics results indicated that velocity magnitude, vorticity magnitude, viscous dissipation, energy loss, and kinetic energy revealed statistical differences between volunteers and patients, particularly at peak systole and e-wave.

Some of the parameters reported in this study have been reported before. Nevertheless, those parameters have been obtained from different methodologies in different cohorts of patients. In our case, we calculated several parameters from a single segmentation at once from only one 4D flow dataset, which is difficult to determine with other methods.

Some other methods are based on a finite difference approach, as in Lorenz et al. [33]. However, it is well known that finite difference cannot effectively handle complex geometries, such as those found in the cardiovascular system. Neither can impose boundary conditions on irregular surfaces directly but they are both sensitive to noise. Fouras et al. showed that this approach suffers from a loss of accuracy in estimating hemodynamic parameters due to the omission of out-of-plane velocity information [34]. On the other hand, Sotelo et al. demonstrated the convergence and robustness of the finite element method in cardiovascular flow [9,10]. Further, they also showed that the finite element method is both stable and accurate in the presence of noise.

Although DCM mainly affects the systolic function, we evaluated the hemodynamic parameters at systole and diastole, as several papers have shown that diastolic function is also affected by this disease. For instance, Friedberg et al. and Dragulescu et al. reported that diastolic wall-motion abnormalities are prevalent in pediatric DCM. Their presence is associated with diastolic ventricular dysfunction and adverse outcomes [35,36]. Some papers have assessed diastolic function in DCM patients using 4D Flow data [37,38]. They have described alterations in the flow components related to velocity, vorticity, and kinetic energy in a different cohort of patients, consistent with our results [21–25].

As we showed in this study, intraventricular flow in DCM patients was altered compared to healthy volunteers at diastole. In addition, it is interesting to observe that, while end-diastolic volume was significantly larger in patients than healthy subjects, the maximum hemodynamic values for e-wave and end-diastole were smaller in patients than in volunteers. For instance, we found that in the normal LV, kinetic energy values were high. The highest kinetic energy values were observed during early diastole and regionally distributed near basal LV regions. In contrast, early- and end-diastole kinetic energy was lower than normal in a heterogeneous group of DCM patients and decreased with the LV volume. As we found in our results, this decrease of kinetic energy throughout diastole is associated with viscous dissipation and energy loss. That agrees with a previous study, where comparisons of inflow characteristics in healthy subjects and DCM patients showed more differences at e-wave between the two groups [36–41]. These ventricular diastolic function aspects can be influenced by dynamic load and contractility; these may vary within the spectrum of normal conditions. Furthermore, large ventricles lead to weak suction pumps and have high wall tension, which has been previously suggested to cause energy waste and alter vortex ring dynamics [22,23]. These results indicate that, despite the complex nature of ventricular finding factors, clinically useful information regarding left ventricular diastolic function is associated with distinct mitral flow velocity patterns. Therefore, these alterations of blood flow may be a factor in developing systolic and diastolic dysfunctions.

While eight patients showed significantly impaired systolic function at CMR's time, five showed a complete response to treatment. Despite a nearly normalized LV ejection fraction, it showed similar hemodynamic values to non-responders DCM patients that were markedly different from volunteers. These results suggest a significant increase in the ratio of outflow to inflow during systole in responders DCM patients, but the volumes

were significantly smaller. Therefore, LV ejection fraction cannot reflect subtle ventricular dysfunction, which potentially can be better assessed using flow-based parameters because of the sensitivity to abnormal pumping function [36]. Therefore, the problem of using LV ejection fraction as the pivotal risk marker for DCM patients is that this single parameter does not recapitulate the complexity of the disease.

The location and extent of the changes in intraventricular blood flow, for example, the depth (base to apex) of the vorticity changes or the spread of impaired flow through the ventricular cavity, can potentially be a sensitive marker of the severity of diseases or the progress of the treatment but are hard to quantify because of the 3D nature of the flow. We proposed to use the bullseye plots to depict this data. These plots allow us to display the most important regional differences and extend flow changes in a familiar way to many clinicians. These results could facilitate homogeneity among 4D Flow quantifiable analysis for clinical researchers and clinicians.

The sensitivity study showed a significant relative error, particularly in helicity density, when the differences in the segmentations were greater than 1 pixel in dilation and erosion cases. We performed this sensitivity study even under the pixel resolution of the 4D Flow data. In each pixel, there were four or five elements from the mesh, whose flow values were interpolated from neighborhood pixels. When the segmentation error was lower or equal to 1 pixel, the maximum mean relative error was less than 10% in most hemodynamic parameters studied. Previous research has shown that DCM patients have a lower mean value of velocity magnitude than healthy volunteers [21–25]. These results were considered when the segmentation contour fell inside the LV blood pool. In general, we also observed that lower errors were obtained for almost all hemodynamic parameters when the segmentation underestimated the LV volumes.

Inter-observer and intra-observer assessments showed excellent reproducibility of the results with negligible mean differences and small limits of an agreement at peak systole, e-wave, and end-diastole for all the parameters assessed. It is important to note that the high intra- e inter-observer variability was obtained because we performed an automatic segmentation process using the software Segment. Therefore, the difference in segmentations was minimal, as previously reported by Tufvesson et al. [29]. The automated process corrections were also minimal, which led to a high intra- and inter-observer variability. On the other hand, the sensitivity study was performed by modifying a reference segmentation in the entire contour by applying erosion or dilatation. This result implies a more significant volume difference concerning the reference, and, as a result, high sensitivity to the segmentation was obtained.

The limited size of this proof-of-concept study did not allow us to investigate the prognostic impact, but this will be our aim in future research. Nevertheless, in this small cohort of patients, we have shown that the velocity, vorticity, kinetic energy, viscous dissipation, and energy loss revealed statistical differences between volunteers and patients. This finding could be relevant to assess changes in a longitudinal study or to study the response to a particular therapy. Additionally, 4D Flow derived parameters showed that, in responding DCM patients, hemodynamics parameters were low, even though they had a recovered ejection fraction. Nevertheless, hierarchical cluster analysis underlined that a moderate correlation may exist between ejection fraction and 4D Flow-based metrics, which needs to be studied further. A clinical study involving more DCM patients should be performed in order to corroborate a prognostic impact—and hence a clinical relevance—of 4D Flow analysis in monitoring DCM patients.

The segmentation of the data need was performed over the multi-slice b-SSFP. Additionally, multi-slice b-SSFP and 4D Flow images need to be registered before analyzing the 4D Flow data. Ideally, the segmentation would be made directly on the 4D Flow data. However, the contrast between the blood pool and myocardium in our 4D Flow data was insufficient to perform accurate segmentation. New sequence developments will likely improve contrast in 4D Flow acquisitions, potentially allowing direct segmentation from the 4D Flow data.

## 5. Conclusions

This study describes a methodology for quantitative evaluation of intraventricular hemodynamics using a single segmentation from a 4D Flow dataset. We demonstrate that velocity, vorticity, viscous dissipation, energy loss, and kinetic energy can characterize changes in intraventricular flow in DCM patients compared to healthy volunteers. Further studies should focus on the impact of different treatments of DCM patients on those parameters. Our evidence shows that, although ejection fraction may be recovered, the hemodynamic parameters remain low.

**Supplementary Materials:** The following are available online at https://www.mdpi.com/article/10.3390/app112311165/s1. Table S1: Equations used to calculate each hemodynamic parameter; Table S2: Mean parameter values across 16 segments of the LV, during peak systole, where the bold type is statistically significant between volunteers and patients ($p < 0.05$). $\upsilon$: Velocity magnitude, $\omega$: vorticity magnitude, $H_d$: helicity density, VD: viscous dissipation, EL: energy loss, and K: kinetic energy; Table S3: Mean parameter values across 16 segments of the LV during e-wave, where the bold type is statistically significant between volunteers and patients ($p < 0.05$). $\upsilon$: Velocity magnitude, $\omega$: vorticity magnitude, $H_d$: helicity density, VD: viscous dissipation, EL: energy loss, and K: kinetic energy; Table S4: Mean parameter values across 16 segments of the LV, during end-diastole, where the bold type is statistically significant between volunteers and patients ($p < 0.05$). $\upsilon$: Velocity magnitude, $\omega$: vorticity magnitude, $H_d$: helicity density, VD: viscous dissipation, EL: energy loss, and K: kinetic energy. Figure S1: ROC-curves for hemodynamic parameters (a–f) in the entire LV cavity of the groups of volunteers and patients at peak-systole, e-wave, and end-diastole; Figure S2: Relative error values of volume (a) and each hemodynamic parameter (b–g), obtained comparing the reference segmentation with segmentations given by erosion or dilation, for each group of volunteers and patients at e-wave. * indicates statistically significant differences ($p < 0.05$); Figure S3: Relative error values of volume (a) and each hemodynamic parameter (b–g), obtained comparing the reference segmentation with segmentations given by erosion or dilation, for each group of volunteers and patients at end-diastole. * indicates statistically significant differences ($p < 0.05$); Figure S4: Bullseye plots of mean hemodynamic parameters (a–f) across 16 segments for volunteers (i) and patients (ii) at the e-wave. * indicates statistically significant differences ($p < 0.05$).

**Author Contributions:** All authors were actively involved in reviewing and drafting the manuscript. All authors have approved the final version of this manuscript. P.F. developed the methodology to assess intracardiac flow, process all data, and perform statistical analysis and wrote the manuscript. J.S. developed the finite element method to evaluate the hemodynamic parameters and edited the manuscript. C.M. was the second observer in the inter-observer analysis and edited the manuscript. B.R. participated in the recruitment of patients, acquired CMR images, and edited the manuscript. E.K. developed the Eidolon software to register b-SSFP data and 4D Flow for the cohort of DCM patients and edited the manuscript. D.N. participated in the design of the study to recruit patients and edited the manuscript. J.M. participated helped develop the finite element method to assess hemodynamic parameters and edited the manuscript. D.H. participated in the development of the methodology to assess intracardiac hemodynamics and edited the manuscript. S.U. participated in the study's design, developing the method to assess intracardiac flow, helping with statistical analysis, and wrote the manuscript. All authors have read and agreed to the published version of the manuscript.

**Funding:** This work has been funded by projects PIA-ACT 192064 and the Millennium Nucleus on Cardiovascular Magnetic Resonance NCN17_129 of the Millennium Science Initiative, both of the National Agency for Research and Development, ANID. The authors also thanks to Fondecyt project 1181057 also by ANID. Franco P. thanks to ANID-PCHA/Doctorado-Nacional/2018-21180391. Sotelo J. thanks to CONICYT-FONDECYT Postdoctorado 2017 #3170737 and ANID-FONDECYT de Iniciación en Investigación #11200481.

**Institutional Review Board Statement:** All subjects participated under informed consent, with data collection approved by the Regional Ethics Committee, South East London, UK (REC, 12/LO/1456).

**Informed Consent Statement:** Informed consent was obtained from all subjects involved in the study.

**Data Availability Statement:** The datasets generated during and/or analyzed during the current study are not publicly available due to data privacy according to the rules of King's College London but could be available from Bram Ruijsink (b.ruijsink@gmail.com) on reasonable request.

**Acknowledgments:** This work has been funded by projects PIA-ACT 192064 and the Millennium Nucleus on Cardiovascular Magnetic Resonance NCN17_129 of the Millennium Science Initiative, both of the National Agency for Research and Development, ANID. The authors also give thanks to Fondecyt project 1181057, also by ANID. Franco P. gives thanks to ANID-PCHA/Doctorado-Nacional/2018-21180391. Sotelo J. gives thanks to CONICYT-FONDECYT Postdoctorado 2017 #3170737 and ANID-FONDECYT de Iniciación en Investigación #11200481.

**Conflicts of Interest:** The authors declare no conflict of interest.

## Appendix A. Intra- and Inter-Observer Reproducibility

As shown in Figure A1, excellent intra-observer agreement with minimal mean differences and small limits of agreement were found for peak systole. Mean differences were: velocity magnitude $-0.0003 \pm 0.0118$ m/s, kinetic energy $(-0.4580 \pm 0.7490) \times 10^{-9}$ J, vorticity magnitude $0.0029 \pm 0.0249$ 1/s, helicity density $(-0.0978 \pm 0.5345) \times 10^{-3}$ m/s$^2$, viscous dissipation $-0.0070 \pm 0.0578$ 1/s$^2$, and energy loss $(-0.0124 \pm 0.2563) \times 10^{-9}$ W. Similar results were obtained at e-wave and end-diastole, as shown in Figures A2 and A3, respectively. Figure A4 demonstrates excellent inter-observer analysis agreement for peak systole. Mean differences were: velocity magnitude $-0.0024 \pm 0.0124$ m/s, kinetic energy $(-0.0349 \pm 0.1093) \times 10^{-5}$ J, vorticity magnitude $0.1122 \pm 0.7634$ 1/s, helicity density $-0.0001 \pm 0.0139$ m/s$^2$, viscous dissipation $-2.1652 \pm 11.8205$ 1/s$^2$, and energy loss $(0.0012 \pm 0.3008) \times 10^{-8}$ W. Figures A5 and A6 show the results obtained at e-wave and end-diastole, respectively, with comparable results at peak systole.

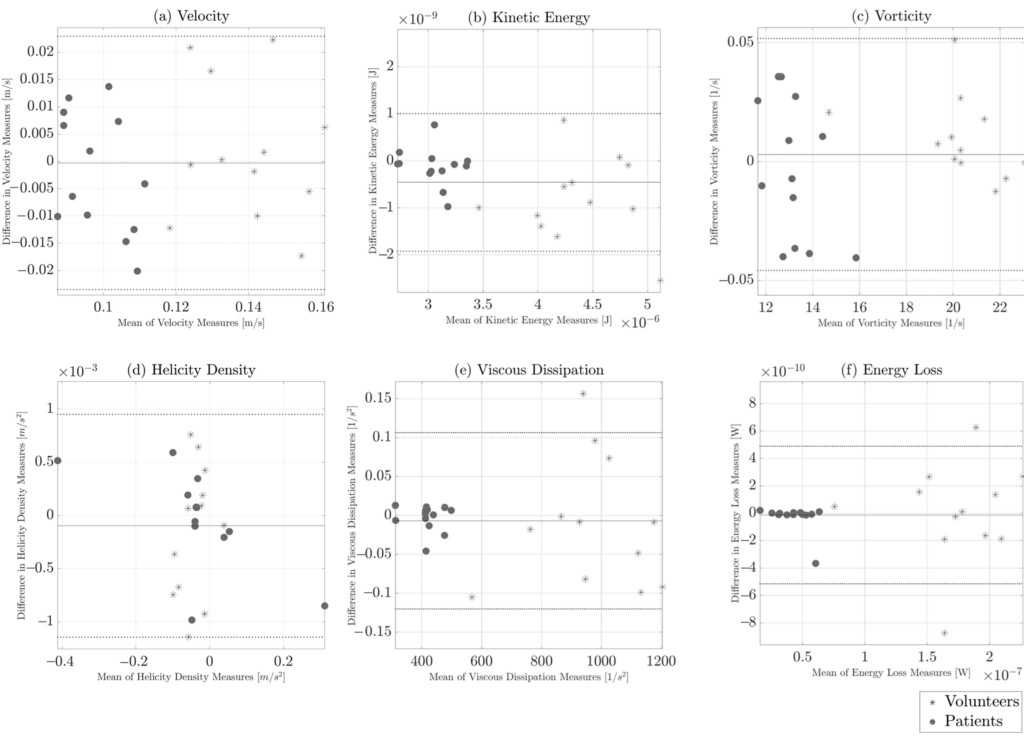

**Figure A1.** Bland-Altman plots represent the intra-observer reproducibility in the measurements of LV global hemodynamic parameters (**a**–**f**) at peak systole. The thick line represents the mean difference, and the thin lines represent the limits agreement (1.96 SD).

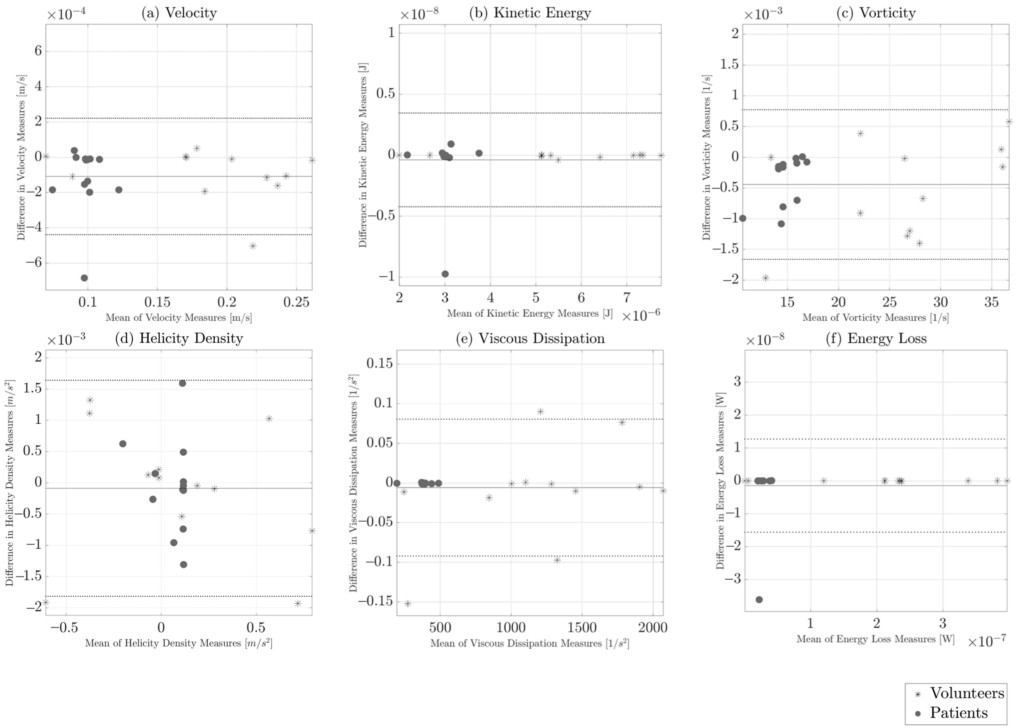

**Figure A2.** Bland-Altman plots represent the intra-observer reproducibility in the measurements of LV global hemodynamic parameters (**a**–**f**) at e-wave. The thick line represents the mean difference, and the thin lines represent the limits agreement (1.96 SD).

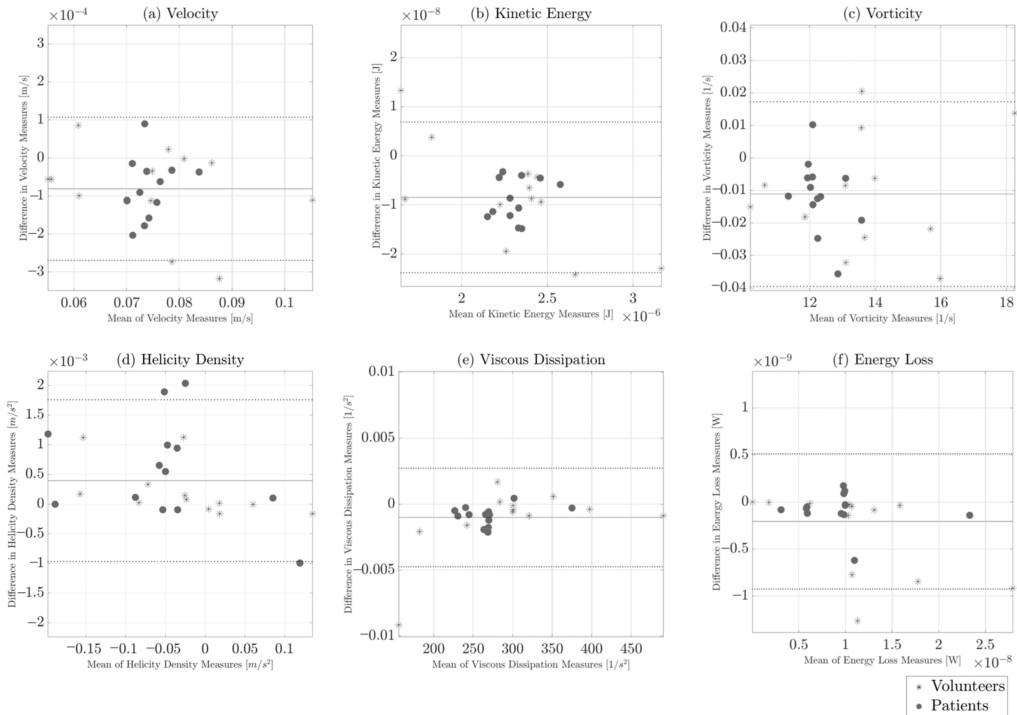

**Figure A3.** Bland-Altman plots represent the intra-observer reproducibility in the measurements of LV global hemodynamic parameters (**a**–**f**) at end-diastole. The thick line represents the mean difference, and the thin lines represent the limits agreement (1.96 SD).

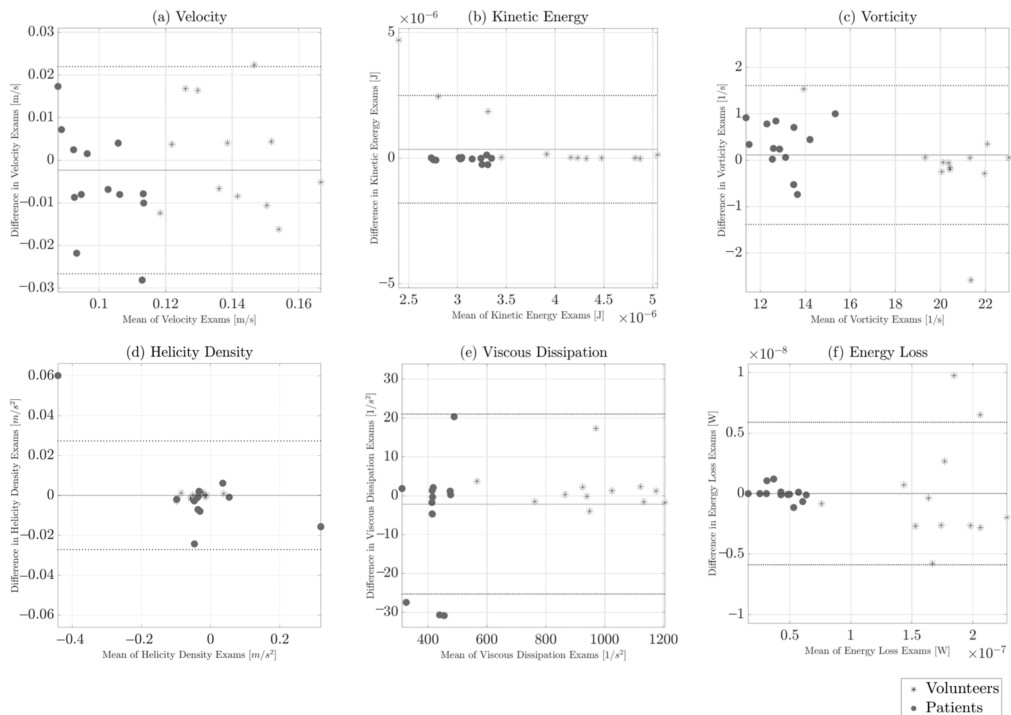

**Figure A4.** Bland-Altman plots represent the inter-observer reproducibility in the exams of LV global hemodynamic parameters (**a**–**f**) at peak systole. The thick line represents the mean difference, and the thin lines represent the limits agreement (1.96 SD).

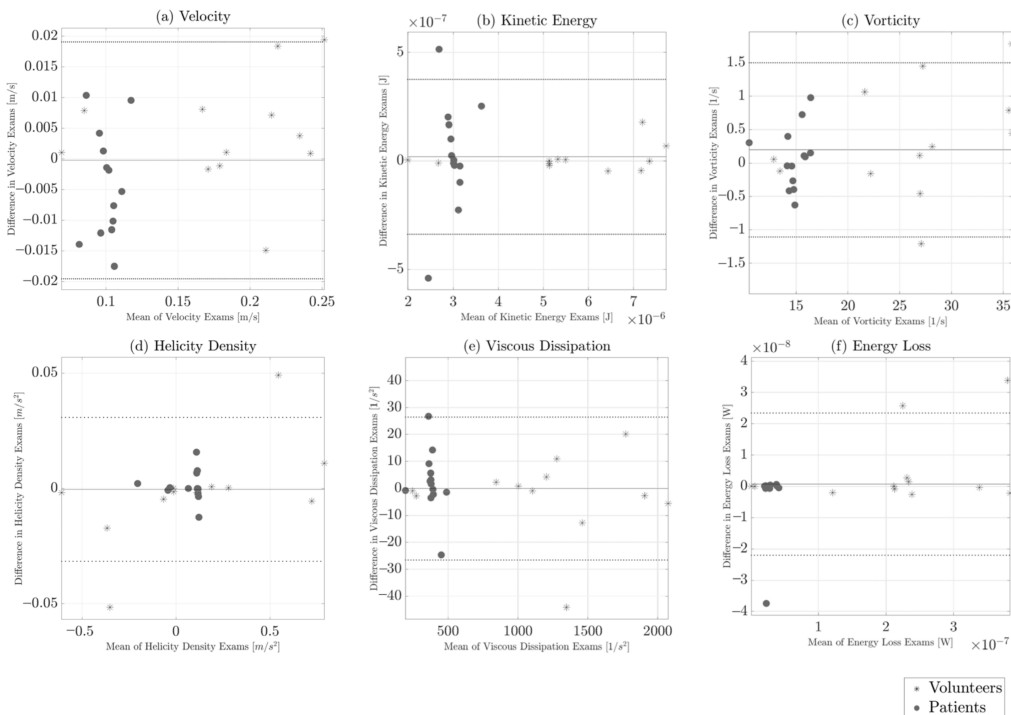

**Figure A5.** Bland-Altman plots represent the inter-observer reproducibility in the exams of LV global hemodynamic parameters (**a**–**f**) at e-wave. The thick line represents the mean difference, and the thin lines represent the limits agreement (1.96 SD).

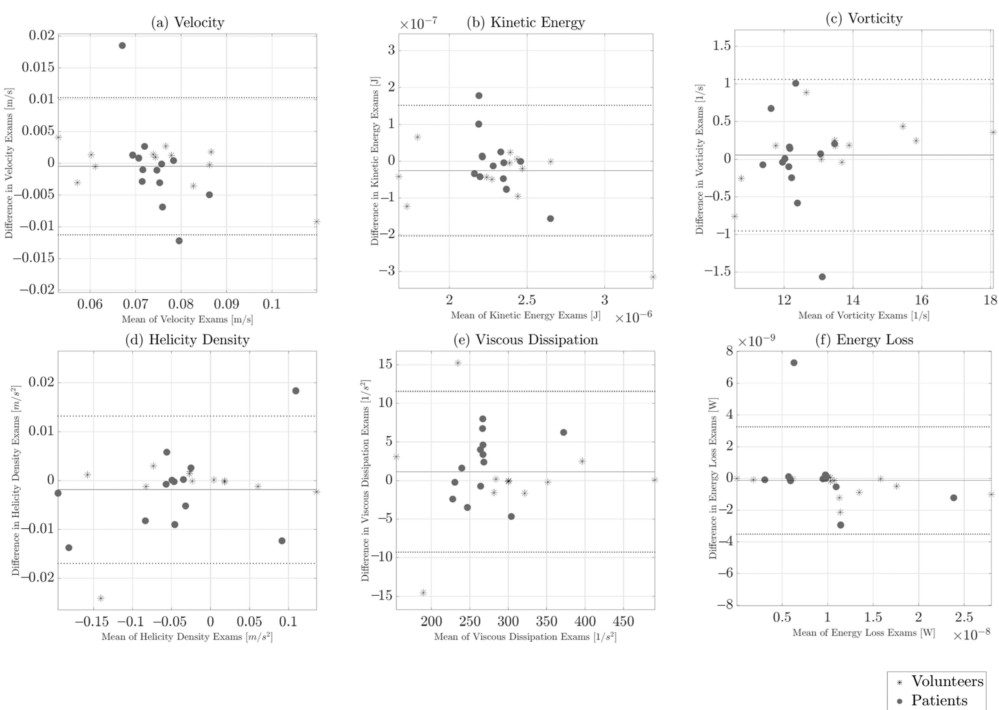

**Figure A6.** Bland-Altman plots represent the inter-observer reproducibility in the exams of LV global hemodynamic parameters (**a**–**f**) at end-diastole. The thick line represents the mean difference, and the thin lines represent the limits agreement (1.96 SD).

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
