# Peer review of "Comprehensive Assessment of Left Intraventricular Hemodynamics Using a Finite Element Method: An Application to Dilated Cardiomyopathy Patients"

_applsci, doi:10.3390/app112311165_

Round 1

Reviewer 1 Report

I read with interest the work proposed by Franco and colleagues.

The study, well designed and organized, proposed a methodology for the quantitative assessment of LV hemodynamics using a single segmentation from a 4D Flow dataset.

The following major concerns should be addressed:

  1. The use of a “single” segmentation, which is part of the novelty of the work, requires further clarifications. It is not clear if a single static segmentation was reconstructed and used (in case, at which phase?) or if the single segmentation was then dynamically adapted over the cardiac cycle. The corresponding paragraph (lines 120-130) in the methods section could be clarified/expanded.
  2. As outlined in the title, a finite element method was employed to improve intracavitary analysis. However, the authors did not underline the real advantage/benefit of using such an approach to improve 4D Flow analysis if compared to the conventional strategy.
  3. The current proof-of-concept was aimed at confirming the alteration of LV hemodynamics in DCM patients when compared to healthy subjects. Though tested on a small cohort of patients, hierarchical cluster analysis underlined that correlation may exist between ejection fraction (EF) and 4D Flow-based metrics. However, in order to corroborate a prognostic impact – and hence a clinical relevance - of 4D Flow analysis in monitoring DCM patients, 4D Flow-derived metrics should be expected to outperform the clinical relevance of conventional clinically-relevant metrics such as EF. This aspect may merit further discussion.

Minor comments

Line 95: please clarify which treatment was received by DCM patients.

Lines 71-76: this is a relevant paragraph but simplifying or splitting it could help the reader.

Line 150: the definition of LV centerline is not clear.

Grammar check is suggested for minor language and style corrections.

Reviewer 2 Report

The manuscript presents a comprehensive assessment of left intraventricular hemodynamics for cardiomyopathy patients. The manuscript is well written and technically sound with comprehensive results and discussions.

I recommend the manuscript for publication, pending minor checking of the typos and spell checks, such as at line 29 "ere"->"were". Please check throughout the manuscript.
